# Molecular insights into degron recognition by CRL5[ASB7] ubiquitin ligase

Mengyu Zhou[1,5], Xiaolu Wang[2,5], Jiangtao Li[1], Jinfeng Ma[1], Ziyu Bao[3], Xiaojie Yan [1], Bing Zhang [1], Tong Liu[4], Ying Yu [2] ✉, Wenyi Mi [3] ✉ & Cheng Dong [1,4] ✉

The ankyrin (ANK) SOCS box (ASB) family, encompassing ASB1–18, is the largest group of substrate receptors of cullin 5 Ring E3 ubiquitin ligase. Nonetheless, the mechanism of substrate recognition by ASB family proteins has remained largely elusive. Here we present the crystal structure of ASB7-Elongin B-Elongin C ternary complex bound to a conserved helical degron. ASB7 employs its ANK3-6 to form an extended groove, effectively interacting with the internal α-helix-degron through a network of side-chain-mediated electrostatic and hydrophobic interactions. Our structural findings, combined with biochemical and cellular analyses, identify the key residues of the degron motif and ASB7 required for their recognition. This will facilitate the identification of additional physiological substrates of ASB7 by providing a defined degron motif for screening. Furthermore, the structural insights provide a basis for the rational design of compounds that can specifically target ASB7 by disrupting its interaction with its cognate degron.

Ubiquitin (Ub)-proteasome system (UPS) is the predominant protein degradation pathway governing over 80% of intracellular proteolysis[1]. In this pathway, the target proteins are covalently attached to Ub through the sequential action of a three-enzyme cascade of Ub-activating enzyme (E1), Ub-conjugating enzyme (E2), and Ub ligase (E3), and subsequently recognized and degraded by the 26 S proteasome[2]. E3 ligases are key regulators in this system as they determine the specificities for ubiquitination and degradation of the substrates by recognizing a specific motif (called degron) of the substrate[3,4]. The human genome encodes more than 600 E3 ligases, which can be divided into two major classes based on the mechanism of ubiquitin conjugation: HECT (homologous to E6-AP C-terminus) domain-containing E3s, and RING (really interesting new gene) domain-containing E3s[5]. In the case of HECT E3s, Ub is transferred sequentially from E2 to the HECT domain and then to the substrate[6], whereas for the RING E3s, Ub is transferred directly from E2 to the substrate[7].

Multi-subunit cullin-RING ligases (CRLs) are the largest family of E3s[8], responsible for ~20% of UPS substrate turnover[9] and involved in numerous cellular processes[10]. CRLs utilize a cullin scaffold protein and a catalytic RING subunit (Rbx1 or Rbx2) to constitute the core subunit while employing distinct adaptors and substrate receptors to assemble unique E3 complexes[11]. The cullin family consists of six canonical members (Cul1, Cul2, Cul3, Cul4A, Cul4B, and Cul5)[12], along with one atypical Cul7[13]. In particular, Cul1 uses Skp1 and F-box proteins as the adaptor and substrate receptors, respectively[14]. Cul4 employs DDB1 and DCAF proteins as the adaptor and substrate receptors, respectively[12,15]. Cul3 is unusual in that it lacks an adaptor protein and uses BTB-domain proteins as the substrate receptors[16]. Cul2 and Cul5 both use Elongin B/C as the adaptor, while employing a distinct subfamily of BC-box proteins as substrate receptors[17]. Specifically, in the case of Cul5, SOCS (suppressors of cytokine signaling)-box proteins function as the substrate receptors, including SH2,

[1]The Province and Ministry Co-sponsored Collaborative Innovation Center for Medical Epigenetics, Key Laboratory of Immune Microenvironment and Disease (Ministry of Education), Department of Biochemistry and Molecular Biology, School of Basic Medical Sciences, Tianjin Medical University, Tianjin 300070, China. [2]Department of Pharmacology, Tianjin Key Laboratory of Inflammatory Biology, Center for Cardiovascular Diseases, Tianjin Medical University, Tianjin 300070, China. [3]Tianjin Institute of Immunology, Department of Immunology, School of Basic Medical Sciences, Tianjin Medical University, Tianjin 300070, China. [4]Department of Cardiology, Tianjin Institute of Cardiology, Second Hospital of Tianjin Medical University, Tianjin 300211, China. [5]These authors contributed equally: Mengyu Zhou, Xiaolu Wang. ✉e-mail: yuying@tmu.edu.cn; wenyi.mi@tmu.edu.cn; dongcheng@tmu.edu.cn

ankyrin (ANK) repeats, WD40 repeats, SPRY domain-containing proteins[18–20].

The ankyrin SOCS box (ASB) family comprising ASB1–18 represents the largest class of SOCS box proteins and has been implicated in numerous signaling pathways[21,22], but little is known about their E3 ligase activity and degradation substrates[23]. Among them, ASB7, functions as a regulator of cytoskeletal organization[24], endoplasmic reticulum stress response[25], and cell division[26]. ASB7 has been identified to target physiological substrate DDA3 for proteasomal degradation[26], and recent global protein stability (GPS) analysis unveiled its role in recognizing internal degrons of multiple substrates[27]. The binding model between ASB7 and its degron, predicted by AlphaFold2, has greatly enhanced our understanding of the molecular mechanisms[27]. However, structurally validated by experimentally determined structure still warrants further investigation. In this study, we report the crystal structure of ASB7-ElongB/C in complex with its cognate degron, shedding light on the detailed recognition of the internal degron by the ASB7. This structural insight provides valuable information for understanding the substrate specificity of ASB7 and facilitates the development of chemical probes against ASB family proteins.

## Results

### ASB7 binds its cognate degrons in vitro

ASB7 functions as the substrate receptor of CRL5 ubiquitin ligase complex, comprised of Elongin B, Elongin C, Cul5, and Rbx2 subunits[21]. ASB7 contains eight ANK repeats followed by C-terminal BC and Cul5 boxes that mediate the Elongin B/C and Cul5 binding, respectively. Previous studies have identified ASB7's role in targeting an internal degron for degradation[27]. To explore the direct binding ability of ASB7 to degrons in vitro, we performed glutathione S-transferase (GST) pull-down assays (Fig. 1a). Five distinct degrons, derived from the reported substrate proteins[27] CLIP2, GIGYF1, CCDC17, CEP152 and LZTS1, each comprising an ~36-residue sequence, were individually fused to the C-terminus of GST as bait. Nearly the full-length human ASB7 (residues 11–318) was co-expressed with Elongins B and C, as ASB7 is unstable when expressed alone. The ternary complex was subsequently purified from *Escherichia coli*. The results showed that all the GST-fusion

degrons can directly bind to ASB7. Moreover, ASB7-Elongin B-Elongin C ternary complex can form a stable quaternary complex with the degron analyzed by size exclusion chromatography (Supplementary Fig. 1a).

To further characterize the interaction of ASB7 with its degron, we constructed a series of truncated mutants of ASB7 containing different number of ANK repeats and performed GST pull-down assay (Supplementary Fig. 1b, c). We found that the deletion of ANK1–2 did not impede the interaction with the LZTS1-degron. In contrast, more extensive deletions, such as the removal of ANK3 or ANK4, resulted in the inability to pull down LZTS1-degron, indicating the significance of ANK3–8 rather than the N-terminal ANK1–2 in mediating the interaction with the degron.

For a more precise quantification of this interaction, we synthesized corresponding 23-residue peptides and carried out isothermal titration calorimetry (ITC) measurements (Fig. 1b–f and Supplementary Table 1). Consistent with GST pull-down assays, ASB7 exhibited strong binding to all five degrons. Notably, CLIP2-, GIGYF1-, CCDC17-, and LZTS1-degrons displayed comparable binding affinities, with a dissociation constant ($K_D$) ranging from 0.7 to 1.1 µM, whereas CEP152-degron exhibited slightly weaker binding, with a $K_D$ value of 3.8 µM. Taken together, these results suggest that the ASB7 can directly bind to its degrons in vitro, with a low-micromolar affinity, and ANK3–8 is required for degron-binding.

### Structure of ASB7 bound to the LZTS1-degron

To investigate the recognition mode of degrons by ASB7, we determined the crystal structure of ASB7-Elongin B-Elongin C complex bound to a 23-residue peptide derived from LZTS1 substrate. Data collection and structure refinement statistics are summarized in Table 1. The structure shows that the eight tandem ANK (ANK1–8) repeats of ASB7 stack together to form an elongated conformation, while the SOCS box assembles with Elongin B/C into a global structure suspended at the C-terminus of ASB7 (Fig. 2a). Each ANK repeat folds into a canonical helix-loop-helix structure with a hairpin loop projecting at a 90° angle away from the helical pairs, forming an L-shaped cross-section[28,29]. However, ANK6 has a shorter hairpin loop compared to the others, leaving a gap between the hairpin loops of ANK5 and

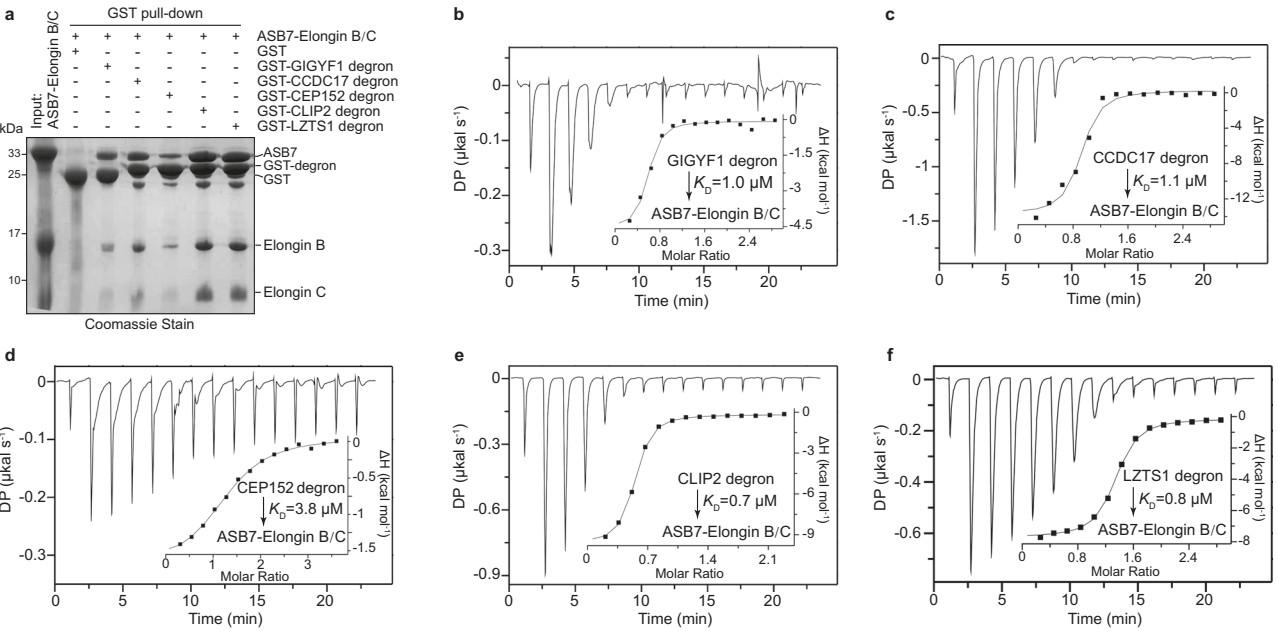

**Fig. 1 | ASB7 directly binds to its degrons in vitro. a** GST pull-down assays using GST-tagged GIGYF1, CCDC17, CEP152, CLIP2, and LZTS1 peptides to pull-down purified ASB7-Elongin B/C. Source data are provided as a Source Data file.

Representative image, $n$ = 3. **b–f** ITC titration and binding curves of purified ASB7-Elongin B/C with different degrons. Binding affinities ($K_D$) are indicated. Source data are provided as Source Data file.

ANK7. Importantly, ASB7 creates a substrate binding groove through the concave inner helix and hairpin loops of ANK3–6. This groove, ~30 Å long and ~9 Å wide, is enclosed by ANK3 and ANK6 at both ends (Fig. 2b). The bottom of the groove is lined with hydrophobic residues, whereas the sides of the groove are made up of mixed-charge residues.

**Table 1 | Data collection and refinement statistics**

|  | ASB7-LZTS1-degron |
|---|---|
| PDB accession number | 8Y1U |
| Data collection |  |
| Space group | P 2₁ |
| Cell dimensions |  |
| *a, b, c* (Å) | 48.15, 43.72, 159.23 |
| α, β, γ (°) | 90.00, 92.01, 90.00 |
| Resolution (Å) | 48.21–2.41 (2.50–2.41)ᵃ |
| $R_{sym}$ or $R_{merge}$ | 0.093 (0.846) |
| $I/σI$ | 14.47 (2.17) |
| Completeness (%) | 99.67 (98.70) |
| Redundancy | 6.6 (6.7) |
| Refinement |  |
| Resolution (Å) | 48.12–2.41 (2.50–2.41) |
| No. reflections | 49860 (4433) |
| $R_{work}/R_{free}$ | 0.2251/0.2463 |
| No. atoms |  |
| Protein | 3106 |
| Water | 96 |
| *B*-factors |  |
| Protein | 86.8 |
| Water | 63.9 |
| R.m.s. deviations |  |
| Bond lengths (Å) | 0.008 |
| Bond angles (°) | 1.25 |

ᵃValues in parentheses are for the highest-resolution shell.

The LZTS1-degron adopts an α-helical structure spanning ~5 turns over 18 residues, followed by an extended loop of terminal residues. The α-helix of LZTS1-degron fits snugly into the binding groove of ASB7 in a complementary fashion, with the terminal residues protruding from the groove.

The electron density maps of ANK repeats, and LZTS1-degron are well-defined, although some regions of Elongin B/C subunits exhibit poor electron density. Nevertheless, the overall structure shows that the SOCS box folds into three conserved helices along the periphery of the outer helices of ANK6–8, which in turn contacts with Elongin C to form a four-helix bundle that associates with the C-terminus of Elongin B (Fig. 2a). This architecture resembles that of other SOCS box-Elongin B/C structures[30–32], suggesting a common mechanism of ubiquitination in these cullin-dependent E3 ligases.

## Molecular recognition of the degron by ASB7

The α-helical LZTS1-degron, extending from N terminus to C terminus, aligns in an antiparallel orientation against the ANK repeats from ANK6 to ANK3. We here designated the initial glutamic acid residue of the α-helical degron as position 1 (E1) (Fig. 2c). In this study, we use the one-letter code to denote degron residues and the three-letter code for ASB7 residues (excluding the mutant version) for clarity. Based on the structure, five residues E1-L5-E8-L12-L16 on the α-helical degron are almost entirely buried in the bottom of the groove, defining this region as contact site 1. By contrast, residues L2-M6-Q9-E13-R17 are associated with the solvent-exposed surface on one side of the groove, termed site 2, while residues L11 and K15 on the other side of the groove constitute site3 (Fig. 2d, e).

At site 1, two negatively charged residues E1 and E8 mediate the ASB7 bound through electrostatic bonds. First, the carboxyl group of E1 forms a salt bridge and hydrogen bond with the guanidine group of Arg185 from ANK6 (Fig. 3a). Second, the carboxyl group of E8 engages three hydrogen bonds with the side chains of Thr151, Ser147 and Trp118 (Fig. 3b). In addition, the nonpolar residues L5, L12 and L16 establish extensive hydrophobic interactions with ASB7. Specifically, L5 inserts into a hydrophobic cavity formed by Phe182, Ile159 and Leu156 (Fig. 3c). On the other hand, L12 and L16 are closely packed against each other and against aromatic residues Trp118 and Phe82, respectively (Fig. 3d). Furthermore, they are enveloped by

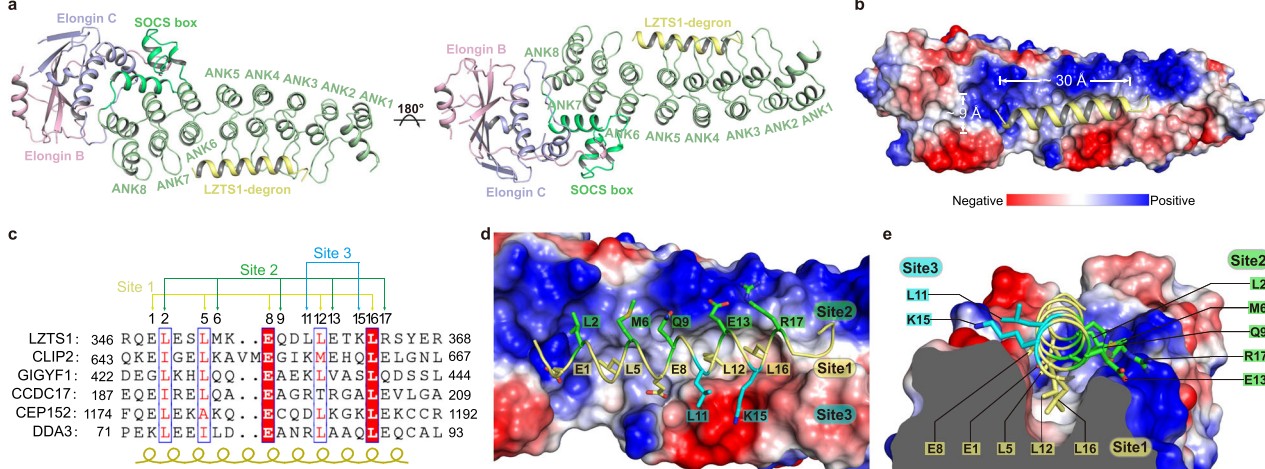

**Fig. 2 | Crystal structure of ASB7-Elongin B/C bound to the LZTS1-degron.**
**a** Overall structure of ASB7-Elongin B/C bound to the LZTS1-degron peptide. ANK repeats, SOCS box, Elongin B and Elongin C are colored in pale green, lime green, light blue, and light pink, respectively. LZTS1-degron is shown in yellow.
**b** Electrostatic potential surface of ASB7 bound to LZTS1-degron. Red, negative; Blue, positive. The helix-binding groove is ~30 Å long and ~9 Å wide. **c** Sequence alignment of ASB7 substrate degrons. The secondary structure and residues involved in site 1, site 2, and site 3 are indicated. **d** The α-helical LZTS1-degron is characterized three sites. Site 1, comprising five residues (yellow), is buried at the hydrophobic bottom of the binding groove. Site 2, comprising five residues (green), resides at the solvent-exposed surface on one side of the positively charged groove. Site 3 (cyan) docks on the other side of the negatively charged groove. **e** Cross-section view of the three different sites.

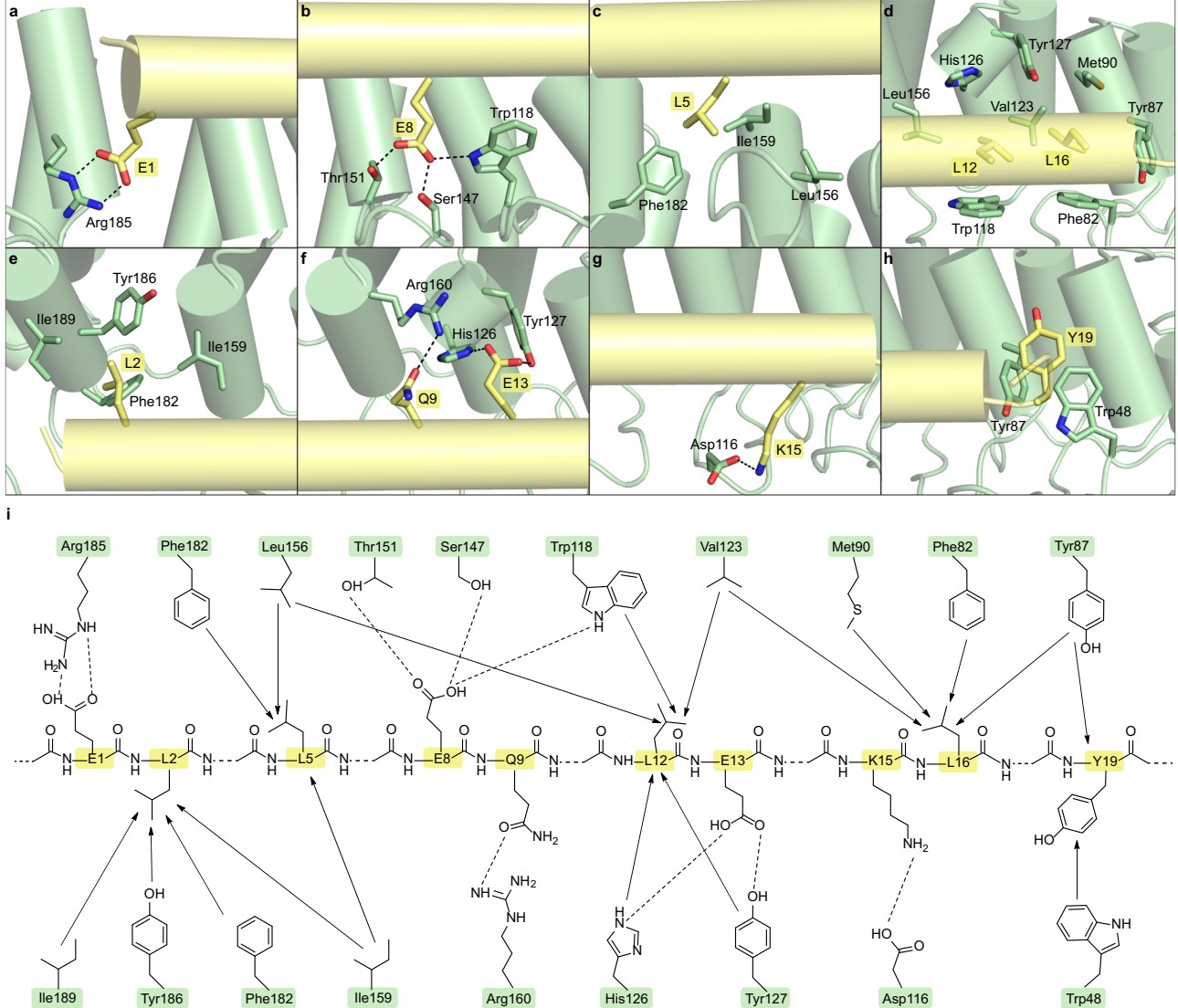

**Fig. 3 | The detailed interactions of ASB7 with LZTS1-degron. a–d** Close-up view of the interactions of E1, E8, L5, L12, and L16 in degron site 1 (yelloworange) with ASB7 (palegreen). The hydrogen bonds are shown as black dashed lines. **e, f** Close-up view of the interactions of L2, Q9, and E13 in site 2 with ASB7. **g** Interactions of K15 in site3 with ASB7. **h** Interactions of the extended Y19 of the LZTS1-degron with ASB7. **i** Schematic of the detailed interactions between ASB7 proteins and LZTS1-degron. Dashed lines and arrows indicate hydrogen bonds and hydrophobic interactions, respectively.

hydrophobic clusters formed by Leu156, Val123, Tyr87, Met90, Tyr127 and His126.

At site 2, the nonpolar L2 is accommodated in a hydrophobic pocket contributed by Ile159, Phe182, Tyr186 and Ile189 of ANK5–6 (Fig. 3e). In contrast, Q9 and E13 are positioned in a basic path, with Q9 forming a hydrogen bond with the guanidine group of Arg160, and E13 fixed by two hydrogen bonds with His126 and Tyr127, respectively (Fig. 3f).

Site 3 exhibits a smaller contact surface compared to site 1 and site 2, partially due to a gap caused by the short hairpin loop of ANK6. Only K15 is anchored to an acidic surface, forming a direct salt bridge with the carboxyl group of Asp116 (Fig. 3g). Unlike the α-helix, the N/C-terminal unstructured segments extend out of the groove and contribute little to ASB7 binding, except for Y19, immediately adjacent to the α-helix C-terminus, which is stabilized by hydrophobic interactions with Tyr87 and Trp48 (Fig. 3h). As a result, the interaction between ASB7 and LZTS1-degron involves a total buried surface area of ~1182 Å².

Collectively, ABS7 predominantly recognizes an α-helix motif through a network of hydrophobic and electrostatic interactions (Fig. 3i). Site 1, containing five residues, serves as the primary binding

site, where two negatively charged residues (E1 and E8) form multiple hydrogen bonds, and three nonpolar leucine residues form robust hydrophobic interactions with the bottom of the binding groove. Site 2 and site 3 further strengthen degron binding, given that their contacts occur on the accessible surface of the complex, which could be accommodated by diverse residue types, indicating a minor contribution to specificity. This observation is mirrored by the results of degron saturation mutagenesis[27].

## Key residues of ASB7 in degron binding

To substantiate the key residues of ASB7 in mediating degron binding, we introduced single-point mutations of ASB7 and assessed their impact on the binding affinities toward the LZTS1-degron peptide by ITC. As expected, in site 1, the substitution of Arg185 with alanine, which would disrupt the electrostatic bonds with E1, led to a 4-fold decrease in binding affinity (Fig. 4a). However, mutations of the E8-interacting Thr151 or Ser147 to alanine severely impaired the degron binding, underscoring the critical roles of these hydrogen bond-mediated interactions. Moreover, alanine mutants of Phe182 diminished the binding affinity by threefold. Of note, replacement of the

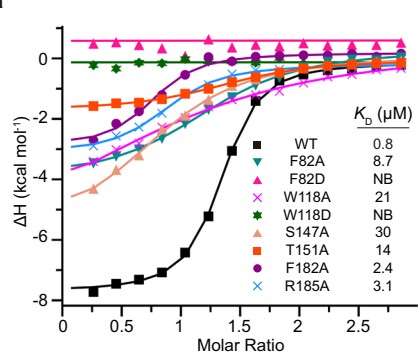

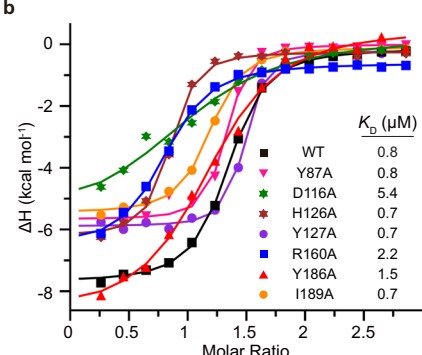

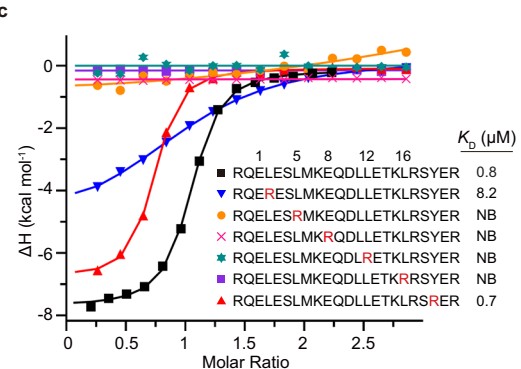

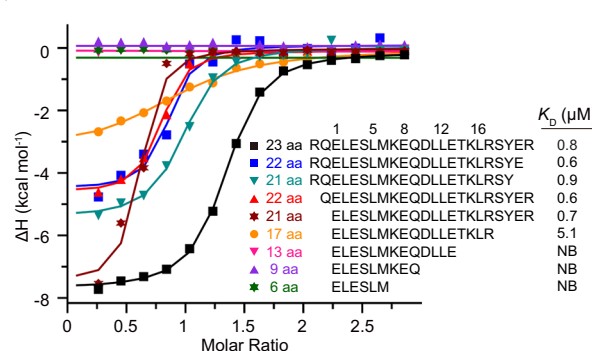

**Fig. 4 | ITC analysis of the key elements required for ASB7 recognition. a** ITC fitting curves of LZTS1 peptide titrated to wild-type ASB7 and site 1 mutants. NB, no apparent binding under our experimental conditions. **b** ITC fitting curves of LZTS1 peptide titrated to wild-type ASB7 and site 2/3 mutants. **c** ITC fitting curves of LZTS1 and substituted peptide titrated against wild-type ASB7. **d** ITC fitting curves of LZTS1 peptides of varying lengths against wild-type ASB7. The corresponding peptide lengths, sequences, and binding affinities are indicated. Source data are provided as Source Data file.

bulky aromatic residue Trp118 in ANK3 or Phe82 in ANK4 with alanine, which is defective in stacking with L12 and L16, respectively, showed a dramatic 10–30-fold decrease in binding affinity, whereas mutating either the aromatic residue with negatively charged aspartic acid resulted in a complete loss of substrate binding, highlighting the importance of these aromatic residues in mediating degron recognition. These results are consistent with the GST pull-down assay, the deletions of ANK3 or ANK4 failed to interact with LZTS1-degron binding (Supplementary Fig. 1b, c).

In site 2, the R160A mutant reduced the binding affinity by threefold, but alanine mutants of L2-interacting Leu189 and Tyr186, or E13-interacting His126 and Tyr127 displayed comparable binding affinities to the wild-type ASB7 (Fig. 4b), suggesting that site 2 makes a minor contribution to substrate recognition. Similarly, the Y87A mutant, attenuating the hydrophobic interaction with Y19, had no effect on the degron binding, consistent with our structural observation that Y19 extends out of the helix-binding groove that could tolerate variant residues (Fig. 3h). Interestingly, the alanine mutant of Asp116, which abolishes the salt bridge with K15 in site3, resulted in a 7-fold reduction in binding, indicating that this acidic path is favored by the positively charged residues embedded in the helix degron.

### Key elements of substrate recognized by ASB7

To explore the essential elements for ASB7 recognition of the α-helix degron, we synthesized a series of mutant peptides in which the site 1 residues were replaced with the basic arginine in an otherwise identical sequence context (Fig. 4c). Our ITC results revealed that arginine substitution of the conserved residues in site 1 including L5R, E8R, L12R and L16R, resulted in an almost complete loss of the ASB7 binding ability. While the L2R mutant in site 2 retained the ability to bind ASB7, albeit with a 10-fold reduced binding affinity. In agreement with the

structural analysis, the Y19R mutant, which extends beyond the groove, exhibited a binding affinity similar to that of the native degron. These findings further corroborate our previous results, emphasizing that site 1 serves as the primary determinant for substrate specificity.

To map the minimal length of the degron required for ASB7 recognition, we examined the binding affinities of ASB7 with peptides of varying length based on the LZTS1-degron (Fig. 4d). The results showed that the deletion of the unstructured region had a minor effect on the ASB7 binding. However, the 17-residue degron exhibited a 6-fold decrease in binding affinity, indicating that ASB7 favors a relatively long helix degron motif for recognition. Further truncation, such as the removal of key nonpolar residue L16, resulted in a complete loss of ASB7 binding.

DDA3, a regulator of spindle dynamics, has been identified as an endogenous substrate targeted by ASB7 for proteasomal degradation to modulate spindle dynamics and genome integrity[26], but how ASB7 recognizes DDA3 remains unknown. By aligning the sequence of DDA3 against the ASB7-targeting degron, we identified the region covering residues 70–93 as a potential degron for ASB7 recognition (Fig. 2c). To test this hypothesis, we synthesized a 23-residue peptide and our ITC result demonstrated that this peptide is able to strongly bind ASB7 with a $K_D$ value of 1.3 μM (Supplementary Fig. 2a). This interaction is further supported by a GST pull-down assay (Supplementary Fig. 2b), indicating that ASB7 recognizes a conserved helical degron motif of substrates for proteasomal degradation.

### ASB7 mediates degron degradation in cells

To verify the role of degron recognition by ASB7 in cells, we carried out a GPS assay, in which the degron was fused to the C-terminus of GFP followed by a 20-residue sequence context devoid of known C-degrons[27]. The stability of the GFP fusion protein was assessed by

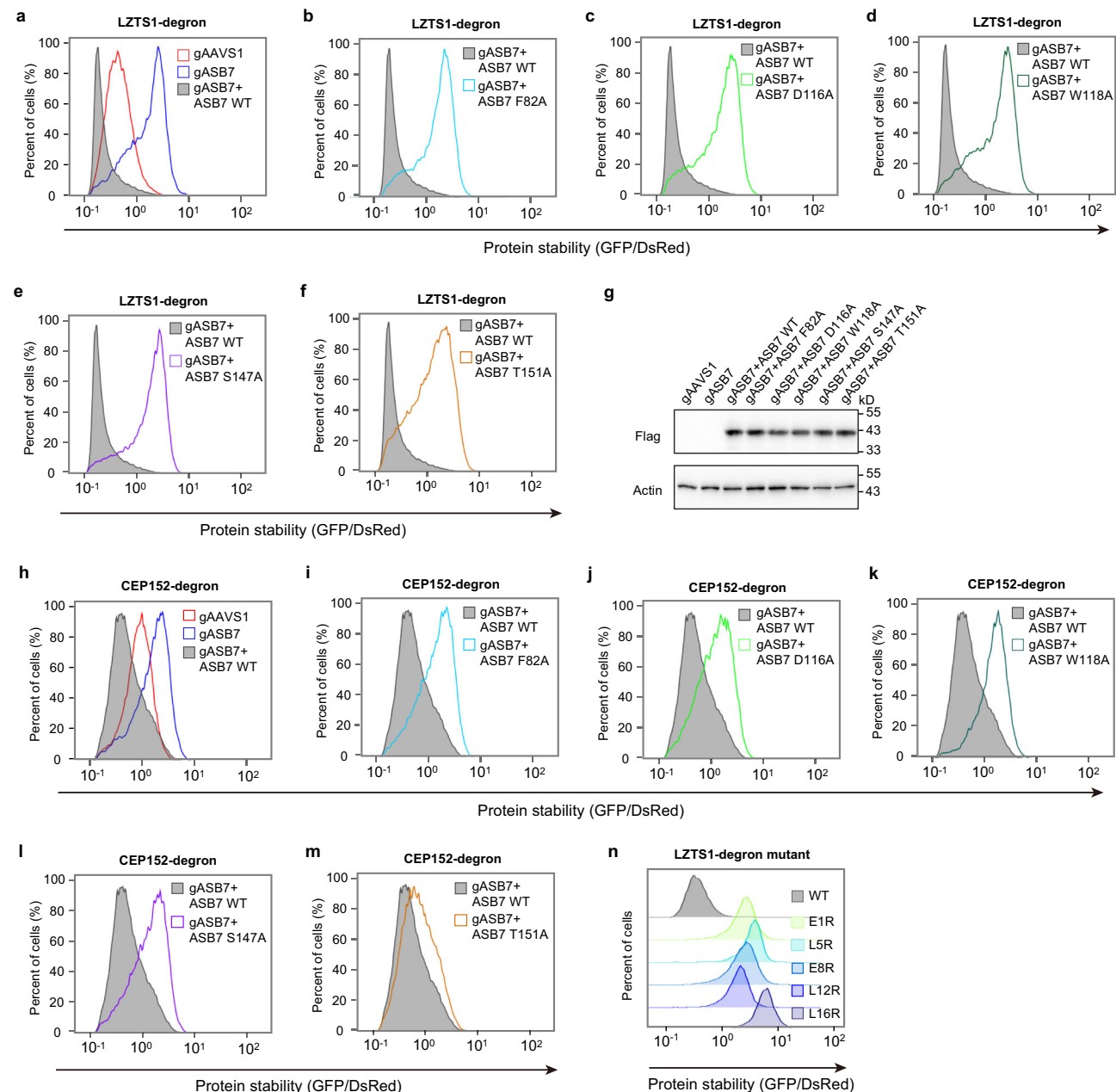

**Fig. 5 | Mutagenesis analysis of ASB7-mediated degradation in cells. a–f** Stability analysis of the GFP-fused LZTS1-degron was performed in ASB7 knockout HEK293T cells with overexpression of exogenous gASB7-resistant wild-type and mutant ASB7. The GFP/DsRed ratio was analyzed by flow cytometry. **g** Western blot analysis of Flag-tagged WT and mutant ASB7 expression in GPS reporter cell lines. Source data are provided as a Source Data file. Representative image, *n* = 3. **h–m** Stability analysis of GFP-fused CEP152-degron with overexpression of exogenous gASB7-resistant wild-type and mutant ASB7 in ASB7 knockout HEK293T cells. **n** Stability comparison of GFP-fused LZTS1-degron with substitutions in contact site 1, monitored by global protein stability assay. Source data are provided as Source Data file.

monitoring the GFP/DsRed ratio using DsRed as an internal reference. The results showed that overexpression of exogenous wild-type ASB7 promoted the degradation of the GFP-fused LZTS1-degron in HEK293T cells (Supplementary Fig. 3a and 5). In contrast, the introduction of the mutants F82A, D116A, W118A, S147A, and T151A, which are defective in degron-binding by ITC, resulted in attenuated LZTS1 degradation (Supplementary Fig. 3b–f). As a control, the Y186 mutant, which retains the strong binding ability to ASB7, can also trigger the degradation (Supplementary Fig. 3g). We further extend this notion by evaluating the mutant's effect on the degradation of CEP152-degron (Supplementary Fig. 3h–o), the results showed a similar profile to that observed with the LZTS1-degron. Additionally, we found that the

defined DDA3-degron can be readily degraded upon overexpression of ASB7 in cells (Supplementary Fig. 2c), suggesting that ASB7 targets substrates for degradation by a conserved recognition mechanism.

To further corroborate the importance of key residues of ASB7 in mediating degron recognition, we conducted "rescue" experiments by ectopically expressing gRNA-resistant wild-type ASB7 or degron-binding deficient mutants in ASB7 knockout cells. We found that depletion of endogenous ASB7 stabilized LZTS1-degron (Fig. 5a). Importantly, ectopic expression of wild-type ASB7, but not the F82A, D116A, W118A, S147A and T151A mutants, restored the degradation of LZTS1-degron in the ASB7 knockout cells (Fig. 5b–g). Consistently, wild-type ASB7, but not the degron-binding deficient mutants,

rescued the degradation of CEP152-degron in ASB7 knockout cells (Fig. 5h–m).

On the other hand, consistent with our structural and ITC analyses, substitution of E1, L5, E8, L12 or L16 with arginine of LZTS1-degron in site 1, resulted in stabilization of the GFP fusion protein relative to their native counterparts (Fig. 5n), further underscoring the importance of the site 1 in mediating degron binding and degradation.

## Discussion

ASB family, comprising 18 members, plays crucial roles in both normal and pathological development[33]. While these members share a conserved SOCS box assembly with Elongin B/C and Cul5, each ASB protein possesses a variable number of ANK repeats that mediate substrate recognition, suggesting that each ASB may have a unique architecture for substrate recognition. However, the understanding of their substrate recognition mechanism has been limited. To our knowledge, prior to this study, only the structure of substrate-bound ASB9 has been described by cryo-electron microscopy at 4.1 Å, in which the N-terminal region of ASB9 is inserted into the homodimer of CKB (creatine Kinase brain-type) substrate[34]. In this study, we provide a 2.4 Å crystal structure of ASB7-Elongin B-Elongin C-degron quaternary complex. We discovered that ASB7 uses ANK3–6, forming a long groove to specifically recognize an amphipathic α-helical degron. Furthermore, our analysis revealed that site 1, containing residues at positions 1, 5, 8, 12, and 16 of the helical degron, is the principal recognition determinant for ASB7. Leveraging this motif, we identified residues 70–93 of the physiological substrate DDA3 as a potential degron, a hypothesis validated by our ITC, GST pull-down, and GPS assay results. This discovery enhances our understanding of ASB7's role in targeting specific substrates for degradation through the recognition of conserved helical degron motif.

Although the helical degron has been docked onto ASB7 using the AlphaFold2-multimer algorithm[27], our comparative analysis reveals significant disparities between the predicted model and our experimentally determined structure. Despite the superimposition of the AlphaFold2 predicted model with our experimentally determined structure yielded a root mean square deviation of 1.3 Å, notable discrepancies emerge, particularly in the regions of ANK6–8 and the SOCS box (Supplementary Fig. 4a). In our experimentally determined structure, the helical degron occupies the predicted binding groove, but with a deviation of ~8 degrees from the predicted orientation. This discrepancy results in key differences in interaction patterns. Specifically, our analysis reveals that the aromatic ring of ASB7 Trp118 is flipped by 90 degrees in the predicted model, leading to a distinct stacking arrangement with degron residue L12 compared to our experimental findings (Supplementary Fig. 4b). Likewise, ASB7 Phe82 undergoes a 90-degree rotation in the predicted model, resulting in poor stacking with L16 due to increased spatial separation, but instead stacks with degron residue Y19. However, in our experimentally determined structure, the Y19 is flipped 180 degrees to avoid steric hindrance with Phe82 (Supplementary Fig. 4b–d). Remarkably, both Trp118 and Phe82 serve as critical residues for substrate recognition and degradation based on our biochemical and cellular assays. The discrepancy between the predicted and experimentally determined structures underscores the importance of our crystallographic analysis in providing a precise understanding of the ASB7-degron interaction landscape. This nuanced insight not only enriches our understanding of substrate recognition mechanisms but also facilitates the rational design of chemical probes targeting this interaction interface.

The recognition of degrons, which can be located at the extreme N-terminus, C-terminus, or internal positions, is a crucial aspect of substrate specificity for E3 ligases[5,35]. The mechanism underlying the recognition of simple N-degrons and C-degrons has been extensively explored through structural biology, revealing that these E3 ligases generally use a closed binding pocket to engage the extremely terminal residues[36], such as GID4[37,38], ZYG11B/ZER1[39,40], KLHDC2[41], TRIM7[42], Pirh2[43]. It is worth noting that ASB7, despite having a closed binding groove, binds to an internal degron. In the closed groove of ASB7, we observed that the interactions between ASB7 and the degron are mediated by amino acid side chains, in contrast to the typical terminal degron recognition, where backbone-mediated interactions are usually indispensable elements. This unique feature allows the helical degron to be extended at both ends without affecting its binding to ASB7. In other words, this helical degron can exist at any position within the substrates, indicating a position-independent recognition mechanism. This suggests that ASB7 may have the capability to recognize a diverse range of substrates, which remains to be investigated. In conclusion, our studies have shed light on the substrate recognition mechanisms by ASB7, providing valuable information for identifying more physiological substrates and facilitating the design of chemical compounds against ASB7.

## Methods
### Protein expression and purification
The human ASB7 (residues 11–318) was amplified via PCR from a human cDNA library and seamlessly cloned into the pET28-MKH8SUMO vector (Addgene plasmid #79526). Simultaneously, sequences encoding human Elongin B (residues 1–104) and Elongin C (residues 17–112) were cloned into the pCDFDuet-1 vector. ASB7, Elongin B, and Elongin C were co-expressed in *Escherichia coli* BL21 (DE3) cells.

Cultures were grown in the Luria-Bertani medium at 37 °C until reaching an OD600 of -0.8. Protein expression was induced by adding isopropyl β-D-1-thiogalactopyranoside to a final concentration of 0.2 mM, and the cells were further incubated at 18 °C overnight. Harvested cells underwent centrifugation at $6680 \times g$ for 8 minutes at 4 °C and were subsequently lysed by sonication in a lysis buffer (20 mM Tris-HCl pH 7.5, 400 mM NaCl, and 2 mM β-mercaptoethanol).

Recombinant proteins were purified by Ni-NTA columns. The N-terminal SUMO tags were cleaved overnight at 4 °C using TEV protease. Further purification of ASB7-Elongin B/C was conducted via Superdex 200 Increase 10/300 GL column (GE Healthcare) in a buffer containing 20 mM Tris-HCl pH 7.5, 150 mM NaCl, and 1 mM dithiothreitol (DTT) at pH 7.5. Mutant proteins followed the same purification protocol.

### Protein crystallization
Prior to the crystallization experiment, the purified ASB7 protein was incubated with the LZTS1 peptide at a molar ratio of 1:1.5 on ice for 30 min. Crystallization experiments were conducted using the sitting-drop vapor diffusion method at 18 °C by mixing 1 μL of the protein and 1 μL of the reservoir solution. ASB7-Elongin B-Elongin C-LZTS1 complex was successfully crystallized in a solution consisting of 0.1 M DL-malic acid at pH 7.0 and 10% (*w/v*) polyethylene glycol 3350.

### Data collection and structure determination
X-ray diffraction data were collected on the beamline BL19U1 at Shanghai Synchrotron Radiation Facility and processed with XDS[44]. The complex structure was solved by the molecular replacement method implemented in Phenix[45] using the AlphaFold2 predicted ASB7 structure as a search template. Refinement of the structural model was performed by Phenix[45], and manual model rebuilding was carried out using COOT[46]. Structural figures were generated by PyMOL program (https://www.pymol.org).

### GST pull-down assay
The DNA sequences encoding the peptides, namely CCDC17 (GMSRLFGLEQEIRELQAEAGRTRGALEVLGARIQELQAE), LZTS1 (VLQLQ QEKRQLRQELESLMKEQDLLETKLRSYEREKTSFG), CLIP2 (SGPGAQQK EIGELKAVMEGIKMEHQLELGNLQAKHDLET), GIGYF1 (SSAGPPGDLEDD

EGLKHLQQEAEKLVASLQDSSLEEEQFT) and CEP152 (GHCFQELEKAKQ ECQDLKGKLEKCCRHLQHLERKHK) were individually cloned into the pGEX vector. Recombinant GST-fused peptides were expressed in *E. coli* BL21 (DE3) cells. The proteins were purified through standard GST affinity chromatography, followed by gel-filtration using a Superdex 200 Increase 10/300 GL column (GE Healthcare).

For the pull-down of different ASB7 fragments, 1 µg of GST-fused LZTS1-degron or GST alone were incubated with 1 µg of various ASB7 fragments with Elongin B/C in binding buffer (50 mM Tris pH 7.5, 300 mM NaCl and 0.05% NP-40) for 4 hour at 4 °C, followed by incubation with 6 µL GST beads rotating at 4 °C for 1 hour. After washing three times by binding buffer, the GST beads were resuspended in 60 µL SDS sample buffer. The results were analyzed western blot with anti-His (Zsbio, TA-02) and anti-GST (Proteintech, 66001-2-Ig).

For the pull-down of different degrons, ~50 µg of purified GST or GST-fusion peptides in 800 µL binding buffer (20 mM Tris-HCl pH 7.5, 400 mM NaCl, and 2 mM β-mercaptoethanol) were separately incubated with 20 µL of glutathione magnetic beads for 1 hour at 4 °C on a rotating wheel. Following three washes with binding buffer, ~30 µg of purified ASB7-Elongin B/C complex were added to the samples and incubated on a rotating wheel for 50 minutes at 4 °C. After three additional washes with binding buffer, the pulled sample was eluted using elution buffer (20 mM Tris-HCl pH 7.5, 400 mM NaCl and 25 mM glutathione). Subsequently, the eluted samples were separated by a 15% SDS-PAGE gel and stained with Coomassie blue for visualization.

### Isothermal titration calorimetry

ITC assays were performed using a MicroCal PEAQ-ITC instrument (Malvern Panalytical) at 16 °C. Peptides were synthesized by Shanghai Apeptide Co., Ltd. All protein and peptide stocks were prepared in an ITC buffer (20 mM Tris-HCl pH 7.5, and 150 mM NaCl) and subsequently diluted to the desired concentrations (39–156 µM for ASB7 protein and 0.8–4 mM for the peptides, respectively).

In each experiment, 1.5 µL of peptide was titrated into the ASB7 complex solution with 15 injections. The injections were spaced at 90 seconds with a reference power of 15 µcal/s. The ITC data were processed using MicroCal PEAQ-ITC Analysis software.

### Cell culture and viral transduction

Human HEK293T (ATCC CRL-3216) cell line was maintained at 37 °C and 5% $CO_2$ in Dulbecco's modified Eagle's medium (BioInd) supplemented with 10% fetal bovine serum (BioInd) and 1% penicillin/streptomycin (BioInd).

Lentivirus was produced through co-transfection of pMD2.G, pPAX2 (Addgene) and pHAGE, lentiCRISPR v2 or pCDH constructs into HEK293T cells using Liposomal Transfection Reagent (Yeasen) according to the manufacturer's recommendations. After 48 hours of culture, lentiviral particle supernatants were collected, filtered through a 0.45 mm filter, and used to infect target cells in the presence of 8 µg/mL polybrene. After 48 hours, the infected cells were selected with hygromycin (200 µg/mL), puromycin (2 µg/mL), and/or blasticidin (10 µg/mL) for indicated pHAGE/ lentiCRISPR v2/pCDH stable clones for 3–4 days before the following experiments.

### CRISPR-mediated knockout of ASB7 and re-overexpression in knockout cells

For ASB7 knockout, oligonucleotides encoding the sense and antisense strands of gRNAs were synthesized, annealed, and cloned into the lentiCRISPR v2 vector. gASB7 sequence was: GCCAACATCGA CATTCAGAA. The gASB7-resistant cDNA of ASB7 was generated using a site-directed mutagenesis kit (Stratagene). The following primers were used for generating gASB7-resistant cDNA: GGACCACAATGCCAA-TATTGATATTCAGAATGGTTTCC, GGAAACCATTCTGAATATCAATATT GGCATTGTGGTCC.

### GPS assay

The oligonucleotide encoding degrons and/or mutants was cloned into a GPS vector using a seamless cloning method. The GFP-degron fusion protein and DsRed protein were expressed from the same transcript. First, the GPS vectors were packaged into lentivirus that was used to transduce HEK293T cells to obtain GPS reporter cells after hygromycin (200 µg/mL) selection. To examine the degradation of degron, exogenous wild-type ASB7 and indicated mutants lentivirus infected the GPS reporter cells. For rescue experiments, ASB7 was knocked out by the introduction of lentiCRISPR v2 gASB7 construct. Then, flag-tagged gRNA-resistant wild-type and mutant ASB7 were overexpressed in knockout cells followed by hygromycin (200 µg/mL), puromycin (2 µg/mL) and blasticidin (10 µg/mL) selection for 3 days. At last, the stability of GFP-degron was determined by measuring the cellular GFP/DsRed ratio through flow cytometry using DsRed as an internal control. Data were collected by LSR Fortressa instrument (Becton Dickinson), and the GFP to DsRed ratio was analyzed with FlowJo software v.10. Expression of wild-type and mutant ASB7 in GPS reporter cells was examined by western blot with anti-flag (Beyotime, AF519) and anti-beta actin (Proteintech, 66009-1-Ig) used as an internal control.

### Western blot

Total cell extracts were made from cells growing in six-well plates. Cells were lysed in a cold lysis buffer (50 mM Tris, pH 7.4, 250 mM NaCl, 0.5% Triton X100, 10% glycerol, 1 mM DTT) supplemented with protease inhibitor cocktail (Yeasen, 20123ES50) and 1 mM phenylmethylsulfonyl fluoride on ice for 20 min and protein concentration was determined by a standard Bradford assay (Beyotime, P0006). SDS-PAGE buffer was then added to 100 µg of lysate from each sample, following denaturing (5 min, 95 °C). Proteins were separated on freshly made 8–10% acrylamide gels and then electro-transferred onto a PVDF membrane. Membranes were blocked (5% skimmed milk in TBS-T) and incubated overnight (4 °C) with primary antibody and secondary antibody 1:20,000 for 1 hour at room temperature. Finally, incubate the membrane in ECL reagent for 1 min in the dark, and then place the blot on a piece of plastic wrap and image.

## Data availability

Atomic coordinates and structure factors have been deposited in the Protein Data Bank under accession code 8Y1U. All study data are included in the article and/or SI Appendix. Source data are provided in this paper.

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

## Acknowledgements

We thank the staff at beamlines BL19U1 of Shanghai Synchrotron Radiation Facility for assistance in X-ray data collection. This work was supported by National Natural Science Foundation of China grants 32271265 (to C.D.), 32071193 (to C.D.), 82321001 (to C.D.) and 82173000 (to W.M.), National Youth Top-Notch Talent Support Program in China, Tianjin Municipal Science and Technology Commission grant 22JCZDJC00440 (to C.D.), Research Foundation of Tianjin Municipal Education Commission grants 2021ZD036 (to C.D.) and 2022KJ191 (to B.Z.), and Core Facility of Research Center of Basic Medical Sciences at Tianjin Medical University.

## Author contributions

C.D. and W.M. conceptualized the project and designed experiments. M.Z. performed protein expression, purification, and crystallization. M.Z., J.L., and J.M. conducted the ITC assays. X.W. and Z.B. carried out the cellular assays. X.Y. determined the crystal structures. C.D., W.M., Y.Y. T.L., and B.Z. analyzed the data. C.D. wrote the manuscript with critical inputs from all authors.

## Competing interests

The authors declare no competing interests.
