## [Transparent Peer Review file · Nature Communications]

Molecular insights into degron recognition by CRL5ASB7 ubiquitin ligase

Corresponding Author: Dr Cheng Dong

Figures originally included in the author's rebuttal have been redacted from this file.

Version 0:

Reviewer comments:

Reviewer #1

(Remarks to the Author)

Building on the work of Elledge and colleagues, Zhou et al. have solved the structure of the Cul5 substrate adaptor ASB7 in complex with its cognate peptide degron motif. Thus, this study demonstrates that ASB7 binds its degron motif directly in vitro, and the experimental structure will clearly be of value to guide further examination of ASB7 function. However, given that the work is essentially only confirming the accuracy of a previously published AlphaFold model, I am unsure of whether the findings convey sufficient novelty to warrant a high-profile publication in a journal such as Nature Communications.

Major points:

Figure 5 – the validation in cells using the GPS assay would greatly benefit from being performed through the genetic complementation of ASB7 knockout cells. I see no reason why this should be technically challenging for the authors, and thus I would consider this experiment essential prior to publication. If an ASB7 antibody is available, it would be nice to see an ASB7 KO population/clone rescued with approximately endogenous levels of the different exogenous ASB7 mutants.

Minor points:

Abstract – concludes with ‘This study will facilitate the identification of additional physiological substrates and the development of chemical compounds targeting ASB7’ – maybe the authors could expand in the text on how it will facilitate the identification of additional substrates, and the potential rationale/feasibility of pharmacological targeting of ASB7 based on the structural data?

Intro, line 49 – the implication is that E3s only recognize short linear degrons, when there are many examples of E3s targeting large/extended/discontinuous/conformation degrons

Intro, line 78/79 – ‘molecular mechanism remains elusive’ – in my opinion this is an overstatement, as it turns out that the previously published AlphaFold models have in fact already revealed the molecular mechanism with pretty remarkable accuracy

Line 96 – could the authors specify why it was necessary to truncate the N-terminus of ASB7 for structural characterisation, rather than use the full-length protein?

Fig. 2 – I think displaying the structure from multiple angles might better illustrate the structural data?

Fig. 2D-F – maybe the three different ‘sites’ could be better annotated/distinguished in some way across these three images?

Fig. 5 – no x-axis label. Confusingly, the order of the mutants is not consistent between the top and bottom set of panels with the two different substrates. Color choices don't seem optimal – or is there some pattern to the coloring, in which case a key would be helpful?

Fig. 5H – not immediately clear that this is now addressing mutations in the degron motif rather than the ligase

Fig. S3 – maybe a different color scheme would help distinguish the two datasets?

Line 246 – ‘ration’ instead of ratio

Reviewer #2

(Remarks to the Author)

This study reveals the detailed molecular mechanisms of degron recognition by the ASB7 E3 ligase complex and uses a mixture of crystallography, ITC, mutagenesis and cellular GPS data. This is particularly noteworthy as it sheds light into the rather understudied ASB E3 family. It is particularly encouraging to see that the degrons discovered by a proteome-wide cellular screen (Zhang, Z. et al. Mol Cell 2023) validated in vitro with robust experimental evidence. The data published previously in the 2023 Mol Cell paper also including AlphaFold2 modelling that quite accurately predicted the ASB7 structure and helical degron binding mode. Thus, the newly submitted work by Zhou et al is important as experimental validation, but not a breakthrough novel story for this journal.

Other minor correction points:

- Line 78-79, ‘the precise molecular mechanism underlying ASB7-mediated recognition of degrons remains elusive’ This seems an inaccurate statement given that the exact degron sequences have mapped and an AF model proposed previously. Better to state that a model has been proposed but not structurally validated.
- Line 93-94, ‘Five distinct degrons, derived from CLIP2, GIGYF1, CCDC17, CEP152 and LZTS1 substrate proteins’ There’s only limited evidence that these proteins are ASB7 substrates, i.e. can be ubiquitinated by ASB7. Perhaps “reported substrate proteins”.
- Line 273 and table 1, the resolutions stated in the text (2.3 Å), data collection (2.25 Å) and refinement (2.6 Å) are inconsistent. Have the data been truncated during data processing causing these differences? Please correct/explain.
- Figure 5 – authors should label the X axis.
- The GPS raw data measurements should be listed in supplementary
- Are all the ITC measurements $n = 1$, or were any repeats performed? It would be possible for the authors to add more interesting information to their paper such as the fitted enthalpy and entropy contributions in supplementary data. The stoichiometry appears quite varied suggesting the protein or peptide concentrations were not always as expected.
- The authors describe ASB7 residues with 3 letter code, whereas they describe the degron residues in single letter code (i.e. such as Leu5 and L5, respectively). The authors don’t explicitly explain that they are using this nomenclature and should introduce the first sentences with both formats with full explanation. The journal may have separate views on whether this formatting is acceptable.

Reviewer #3

(Remarks to the Author)

Author Rebuttal letter:

Reviewer #1 (Remarks to the Author):

Building on the work of Elledge and colleagues, Zhou et al. have solved the structure of the Cul5 substrate adaptor ASB7 in complex with its cognate peptide degron motif. Thus, this study demonstrates that ASB7 binds its degron motif directly in vitro, and the experimental structure will clearly be of value to guide further examination of ASB7 function. However, given that the work is essentially only confirming the accuracy of a previously published AlphaFold model, I am unsure of whether the findings convey sufficient novelty to warrant a high-profile publication in a journal such as Nature Communications.

Response: We appreciate your acknowledgment of the significance of our work in confirming the direct binding of ASB7 to its cognate peptide degron motif, as demonstrated by the experimental structure solved in this study. While AlphaFold used by Elledge and colleagues can accurately predict protein-protein interactions, it cannot replace experimental structures, especially when it comes to precise conformation of main chains and side chains (Terwilliger et al., 2024). In this case, AlphaFold accurately predicted the pocket where ASB7 binds to the degron. However, some parts of interaction patterns differ from the conformation we obtained experimentally, particularly in the conformation of certain key amino acid side chains in ASB7 and the degron as shown in Fig. S4. Through our experimental structure, we identified additional key amino acids mediating degron interactions and degradation as measured by ITC and GPS assays, such as Phe82, Trp118, Ser147 and Thr151, whereas these positions especially Phe82 and Trp118 exhibited noticeable side chain flips in AlphaFold predictions. The precise positions of these key amino acid side chains obtained through crystallography will provide crucial information for designing small molecule ligands

(Hickey et al., 2024). Many ligand designs are based on simulating the interaction conformations between substrates and proteins, and our experimentally obtained structural conformations will offer accurate and novel information for small molecule design. We believe our experimental validation adds critical confirmation to these predictions and provides valuable insights into ASB7 function.

Major points:

Figure 5 – the validation in cells using the GPS assay would greatly benefit from being performed through the genetic complementation of ASB7 knockout cells. I see no reason why this should be technically challenging for the authors, and thus I would consider this experiment essential prior to publication. If an ASB7 antibody is available, it would be nice to see an ASB7 KO population/clone rescued with approximately endogenous levels of the different exogenous ASB7 mutants.

Response: We thank the reviewer for pointing out this. As per your suggestion we knocked out ASB7 in both LZTS1 and CEP152 degron reporter cell lines, which caused greatly increased GFP/DsRed ratio. Then for rescue experiments, we overexpressed flag-tagged gRNA-resistant wild-type and mutant ASB7, and determined the stability of GFP-degron. Our GPS assays showed that F82A, D116A, W118A, S147A, T151A mutants led to major stabilization of GFP compared to WT (Figure R1), which is in agreement with our previous results. We have added this data in the revised manuscript (Figure 5).

To detect both endogenous and exogenous expression levels of ASB7, we ordered ASB7 antibody from ThermoFisher (Catalogue Number PA5-98848), while our WB result showed that there was no specific endogenous or exogenous ASB7 bands in the rescued samples compared with gAAVS1 and gASB7 samples (Figure R2). Thus we have to use Flag Tag antibody to measure exogenous flag-tagged gRNA-resistant wild-type and mutant ASB7 expression levels (Figure R3). We have added this data in the revised manuscript (Figure 5).

[Redacted]

Fig. R1. Mutagenesis analysis of ASB7-mediated degradation in cells. a-f, Stability analysis of the GFP-fused LZTS1-degron was performed in ASB7 knockout HEK293T cells with overexpression of exogenous gASB7-resistant wild-type and mutant ASB7. The GFP/DsRed ratio was analyzed by flow cytometry. g, Western blot analysis of Flag-tagged WT and mutant ASB7 expression in GPS-reporter cell lines. h-m, Stability analysis of GFP-fused CEP152-degron with overexpression of exogenous gASB7 resistant wild-type and mutant ASB7 in ASB7 knockout HEK293T cells. n, Stability comparison of GFP-fused LZTS1-degron with substitutions in contact site 1, monitored by global protein stability assay.

[Redacted]

Fig. R2. Western blot analysis of endogenous and Flag-tagged exogenous WT and mutant ASB7 expression in ASB7 knock-out 293T GPS-reporter cell lines using ASB7 antibody.

[Redacted]

Fig. R3. Western blot analysis of Flag-tagged exogenous WT and mutant ASB7 expression in ASB7 knock-out 293T GPS-reporter cell lines using Flag Tag antibody.

Minor points:

Abstract – concludes with – This study will facilitate the identification of additional physiological substrates and the development of chemical compounds targeting ASB7 – maybe the authors could expand in the text on how it will facilitate the identification of additional substrates, and the potential rationale/feasibility of pharmacological targeting of ASB7 based on the structural data?

Response: We thank the reviewer for this valuable suggestion. We have expanded the text in the Abstract to provide more detail on how our study will facilitate the identification of additional substrates and the potential for pharmacological targeting of ASB7. The revised sentences now read: –Our structural findings, combined with biochemical and cellular analyses, identified the key residues of the degron motif and ASB7 required for their recognition. This will facilitate the identification of additional physiological substrates of ASB7 by providing a defined degron motif for screening. Furthermore, the structural insights provide a basis for the rational design of compounds that can specifically target ASB7 by disrupting its interaction with its cognate degron. –

Intro, line 49 – the implication is that E3s only recognize short linear degrons, when there are many

examples of E3s targeting large/extended/discontinuous/conformation degrons

Response: We thank the reviewer for pointing this out. We have revised this to recognizing a specific motif.

Intro, line 78/79 molecular mechanism remains elusive in my opinion this is an overstatement, as it turns out that the previously published AlphaFold models have in fact already revealed the molecular mechanism with pretty remarkable accuracy

Response: We thank the reviewer for this insightful comment. We agree that advancements in AlphaFold have significantly contributed to understanding the molecular mechanisms. The revised sentence now reads: "The binding model between ASB7 and its degron, predicted by AlphaFold2, has greatly enhanced our understanding of the molecular mechanisms. However, structurally validated by experimentally determined structure still warrants further investigation."

Line 96 could the authors specify why it was necessary to truncate the N-terminus of ASB7 for structural characterization, rather than use the full-length protein?

Response: We thank the reviewer for this comment. The N-terminus (amino acids 1-10) is predicted to be a disordered region, which may hinder crystallization. Therefore, we used a truncated version starting from the well-defined ANK1 domain (deleting amino acids 1-10) for structural characterization.

Fig. 2 I think displaying the structure from multiple angles might better illustrate the structural data?

Response: We thank the reviewer for the suggestion. We have included an additional figure showing the structure from a 180-degree flipped angle in the revised manuscript.

Fig. 2D-F maybe the three different sites could be better annotated/distinguished in some way across these three images?

Response: We thank the reviewer for the suggestion. We have revised the figure to show the three different 'sites' distinguished by different colors, along with a cross-sectional view to better illustrate these sites.

Fig. 5 no x-axis label. Confusingly, the order of the mutants is not consistent between the top and bottom set of panels with the two different substrates. Color choices don't seem optimal or is there some pattern to the coloring, in which case a key would be helpful?

Response: Thank you for your constructive comments. 1) We have added the missing x-axis label to all relevant panels. 2) The order of the mutants has been made consistent between the top and bottom sets of panels for both substrates. 3) We have revised the color scheme by using grey filling for all the wild-type ASB7 and different colored lines for the mutants to enhance clarity.

Fig. 5H not immediately clear that this is now addressing mutations in the degron motif rather than the ligase

Response: We apologize for the confusion. We have added annotations to clarify that it addresses mutations in the degron motif.

Fig. S3 maybe a different color scheme would help distinguish the two datasets?

Response: We thank the reviewer for the suggestion. We have changed the color scheme to better distinguish the two datasets.

Line 246 ration instead of ratio

Response: We apologize for the oversight. We have corrected this to 'ratio' in the revised manuscript.

Reviewer #2 (Remarks to the Author):

This study reveals the detailed molecular mechanisms of degron recognition by the ASB7 E3 ligase complex and uses a mixture of crystallography, ITC, mutagenesis and cellular GPS data. This is particularly noteworthy as it sheds light into the rather understudied ASB E3 family. It is particularly encouraging to see that the degrons discovered by a proteome-wide cellular screen (Zhang, Z. et al. Mol Cell 2023) validated in vitro with robust experimental evidence. The data published previously in the 2023 Mol Cell paper also including AlphaFold2 modelling that quite accurately predicted the ASB7 structure and helical degron binding mode. Thus, the newly submitted work by Zhou et al is important as experimental validation, but not a breakthrough novel story for this journal.

Response: Please refer to reviewer 1 section.

Other minor correction points:

â€¢ Line 78-79, â€œthe precise molecular mechanism underlying ASB7-mediated recognition of degrons remains elusiveâ€

This seems an inaccurate statement given that the exact degron sequences have mapped and an AF model proposed previously. Better to state that a model has been proposed but not structurally validated.

Response: We thank the reviewer for this insightful comment. We agree that advancements in AlphaFold have significantly contributed to understanding the molecular mechanisms. The revised sentence now reads: "The binding model between ASB7 and its degron, predicted by AlphaFold2, has greatly enhanced our understanding of the molecular mechanisms. However, structurally validated by experimentally determined structure still warrants further investigation."

â€¢ Line 93-94, â€œFive distinct degrons, derived from CLIP2, GIGYF1, CCDC17, CEP152 and LZTS1 substrate proteinsâ€

Thereâ€™s only limited evidence that these proteins are ASB7 substrates, i.e. can be ubiquitinated by ASB7. Perhaps â€œreported substrate proteinsâ€.

Response: We thank the reviewer for this valuable feedback. We have revised the sentence to: "Five distinct degrons, derived from the reported substrate proteins CLIP2, GIGYF1, CCDC17, CEP152, and LZTS1."

â€¢ Line 273 and table 1, the resolutions stated in the text (2.3 Å), data collection (2.25 Å) and refinement (2.6 Å) are inconsistent. Have the data been truncated during data processing causing these differences? Please correct/explain.

Response: We thank the reviewer for pointing this out. Initially, we processed the data collection at a high-resolution cutoff of 2.25 Å with $1/\sigma = 1.0$, and truncated the data during refinement due to potential biased signals at high resolution. To address this issue, we reprocessed the data with a high-resolution cutoff of 2.41 Å with $1/\sigma = 2.17$, and refined the data to the same high resolution. The new structural factors have been deposited in the PDB database.

â€¢ Figure 5 â€œauthors should label the X axis.

Response: We thank the reviewer for pointing this out. We have added the X axis in the revised figures.

â€ The GPS raw data measurements should be listed in supplementary

Response: We thank the reviewer for the comment. We have provided the GPS raw data measurements in the supplementary file.

â€ Are all the ITC measurements $n = 1$, or were any repeats performed? It would be possible for the authors to add more interesting information to their paper such as the fitted enthalpy and entropy contributions in supplementary data. The stoichiometry appears quite varied suggesting the protein or peptide concentrations were not always as expected.

Response: We thank the reviewer for the suggestions. We have repeated all ITC measurements at least three times and have provided the details of binding affinity (KD), enthalpy (ΔH), entropy (ΔS), and stoichiometry (n) in the supplementary file.

â€ The authors describe ASB7 residues with 3 letter code, whereas they describe the degron residues in single letter code (i.e. such as Leu5 and L5, respectively). The authors don't explicitly explain that they are using this nomenclature and should introduce the first sentences with both formats with full explanation. The journal may have separate views on whether this formatting is acceptable.

Response: We thank the reviewer for the suggestion. We have introduced the nomenclature in the first sentence of the revised manuscript as follows: "In this study, we use the one-letter code to denote degron residues and the three-letter code for ASB7 residues (excluding the mutant version) for clarity."

Reviewer #3 (Remarks to the Author):

Response:

Thank you very much for your constructive suggestions on our manuscript. Your suggestions are invaluable and have greatly helped us in revising and improving our work.

References

Hickey, C.M., Digianantonio, K.M., Zimmermann, K., Harbin, A., Quinn, C., Patel, A., Gareiss, P., Chapman, A., Tiberi, B., Dobrodziej, J., et al. (2024). Co-opting the E3 ligase KLHDC2 for targeted protein degradation by small molecules. *Nature structural & molecular biology*.
Terwilliger, T.C., Liebschner, D., Croll, T.I., Williams, C.J., McCoy, A.J., Poon, B.K., Afonine, P.V., Oeffner, R.D., Richardson, J.S., Read, R.J., et al. (2024). AlphaFold predictions are valuable hypotheses and accelerate but do not replace experimental structure determination. *Nature methods* 21, 110â116.

Version 1:

Reviewer comments:

Reviewer #1

(Remarks to the Author)

I commend the authors on the thorough revision of this manuscript, and thus I am happy to recommend publication.

Reviewer #2

(Remarks to the Author)

The authors have addressed my comments

Reviewer #3

(Remarks to the Author)
